# Design and Validation with Influenza A Virus of an Aerosol Transmission Chamber for Ferrets

**DOI:** 10.3390/ijerph16040609

**Published:** 2019-02-19

**Authors:** Nathalie Turgeon, Marie-Ève Hamelin, Daniel Verreault, Ariane Lévesque, Chantal Rhéaume, Julie Carbonneau, Liva Checkmahomed, Matthieu Girard, Guy Boivin, Caroline Duchaine

**Affiliations:** 1Centre de Recherche de l’Institut Universitaire de Cardiologie et de Pneumologie de Québec, 2725 Chemin Ste-Foy, Québec City, QC G1V 4G5, Canada; nathalie.turgeon@criucpq.ulaval.ca; 2Département de Biochimie, de Microbiologie et de Bio-Informatique, Faculté des sciences et de génie, Université Laval, Québec City, QC G1V 0A6, Canada; 3Centre de Recherche du Centre Hospitalier Universitaire de Québec, CHUL, 2705 boulevard Laurier, Québec City, QC G1V 4G2, Canada; marie-eve.hamelin@crchudequebec.ulaval.ca (M.-È.H.); rheaumec@hotmail.com (C.R.); julie.carbonneau@crchudequebec.ulaval.ca (J.C.); liva.checkmahomed@crchudequebec.ulaval.ca (L.C.); guy.boivin@crchudequebec.ulaval.ca (G.B.); 4Research and Development Institute for the Agri-Environment, 2700 rue Einstein, Québec City, QC G1P 3W8, Canada; daniel.verreault@environnement.gouv.qc.ca (D.V.); ariane.levesque@irda.qc.ca (A.L.); matthieu.girard@irda.qc.ca (M.G.)

**Keywords:** bioaerosols, influenza virus, ferret animal model, aerosol chamber

## Abstract

*Background*: The importance of aerosols in the spread of viruses like influenza is still a subject of debate. Indeed, most viruses can also be transmitted through direct contact and droplets. Therefore, the importance of the airborne route in a clinical context is difficult to determine. The aim of this study was to design a chamber system to study the airborne transmission of viruses between ferrets. *Methods*: A system composed of three chambers connected in series, each one housing one ferret and preventing direct contact, was designed. The chambers were designed to house the ferrets for several days and to study the transmission of viruses from an infected (index) ferret to two naïve ferrets via aerosols and droplets or aerosols only. A particle separator was designed that can be used to modulate the size of the particles traveling between the chambers. The chamber system was validated using standard dust as well as with ferrets infected with influenza A virus. *Conclusions*: The 50% efficiency cut-off of the separator could be modulated between a 5-µm and an 8-µm aerodynamic diameter. In the described setup, influenza A virus was transmitted through the aerosol route in two out of three experiments, and through aerosols and droplets in all three experiments.

## 1. Introduction

Several infectious diseases are known to be transmissible through the airborne route, such as tuberculosis and measles. The only known disease transmitted only through the airborne route is tuberculosis, as reported by Roy and Milton [1]. Aerosol transmission of other diseases could be preferential or opportunistic [1]. Therefore, it is difficult to assess the importance (or not) of the airborne route in disease transmission. The mode of transmission of some diseases is ambiguous. The evidence of severe acute respiratory syndrome (SARS) airborne transmission was first assessed by indirect evidences such as modeling and epidemiological studies [2]. Moreover, the possible airborne transmission of non-respiratory diseases, like Norovirus, is a subject of investigation [3,4,5].

For many diseases, dissociating transmission routes such as indirect contact, exposure to large droplets and aerosol transmission through aerosols can be complicated, even in controlled laboratory environments. The World Health Organization considers disease transmission with particles >5 µm as droplets transmission and with particles <5 µm as aerosols transmission [6].

The size of the particles involved in the natural transmission of diseases through the airborne route is hard to establish, especially for viral diseases. In fact, only a few studies have looked at the particle size of airborne viruses that can be found in the environment. Anderson 6 stage cascade impactors, National Institute for Occupational Safety and Health (NIOSH) two-stage bioaerosols cyclone samplers [7,8,9], and Sioutas personal cascade impactors [10,11] have been used in agricultural and hospital settings. In all these studies, viruses were found in all air sample stages, meaning that large particles as well as small particles can carry viruses. More recently, in a laboratory setting, experiments using ferrets and particle impactors of various cut-off sizes demonstrated that influenza virus can be transmitted via droplets (15.3–5 µm) as well as airborne particles (5–1.5 µm) [12].

Information on the infectious state of airborne viruses is sparse [13]. Culture on appropriate cell lines is still the gold standard to assess virus infectivity. However, the culture of airborne viruses faces several challenges: (1) low concentrations of viruses in the air require large air volume sampling to allow detection (meaning extensive air sampling periods or the use of high-flow air samplers); (2) viruses can be damaged during air sampling; (3) environmental contaminants can interfere with virus or host cell growth (bacteria, mold, dust, etc.).

The use of animals in laboratory settings can overcome most of these challenges. The virus source can be a sick human, an infected animal, or an artificially generated aerosol. By exposing animals to airborne viruses, air sampling can be avoided (preventing virus damage) as well as laboratory virus culture bias in detection. As an example, using animals instead of air samplers can lead to the demonstration that airborne viruses can [14] or cannot [15] infect healthy animals and also that airborne viruses can remain (or not) infectious long enough to travel to a new host. Using a sick animal or human as an aerosol source has also demonstrated that a sick subject can emit aerosols that can potentially infect other susceptible hosts [12].

Unfortunately, the exposure of healthy animals to aerosols emitted by another animal over several days cannot be performed in commercially available apparatus settings. Indeed, cages designed for animal aerosol exposure are meant for a few minutes per day exposure and cannot be used for housing animals for several days. In contrast, animal cages designed to house subjects over extended periods of time are not airtight, and therefore provide limited information about airborne transmission. These cages can be used to prevent direct contact between index and healthy animals and can be placed at various distances but cannot control the size of particles traveling between cages.

In this study, we designed, constructed and tested a system composed of three airtight cages to study the transmission of infectious agents between animals through large droplets and through airborne particles. The system can house three ferrets for up to 10–12 days and is designed to prevent direct contact between animals. We designed a particle separator to prevent large droplets transmission between cages. The cage system is under negative pressure, with high-efficiency particulate air (HEPA) filters on the air inlet and outlet for the users’ and environment’s protection. This communication describes the main components of this cage system, the particle separator validation using standard aerosol generators as well as a test trial with ferrets and the influenza virus.

## 2. Materials and Methods

### 2.1. Design

An aerosol transmission chamber was developed to study the transmission of infectious agents between infected (index) and naïve ferrets. It was designed to expose naïve animals to naturally produced infectious aerosols containing either both droplets and aerosols or aerosols only. A total of three ferrets could be housed in three individual cages for 10–12 consecutive days (Figure 1).

All three stainless steel (grade 316 L) cages are identical. The interior dimensions are 864 mm wide by 610 mm deep and 610 mm high for a total volume of 321 L per cage. The cages have perforated grates on the left and right sides as well as a 102-mm-high excreta pan with a perforated lid, which serves as a floor for the animals (Figure 2). The available space for the animals inside the cages is thus 784 mm wide by 610 mm deep by 508 mm in height, which exceeds Canadian and European guidelines for ferret housing (https://www.ccac.ca, https://www.coe.int/). Sampling ports are located between the side grates and the extremities of the cages on the top, back and bottom walls of the cages, thus making the ports inaccessible to the animals. The cages are assembled together with a 7.5-mm rubber seal between cages (Figure 2). The distance between the grates of cages number one and two is 102 mm.

The front panel of each cage is composed of a perforated stainless steel door, which is in direct contact with a transparent polycarbonate door (Figure 3A). The polycarbonate door is sealed shut with a 20-mm-thick rubber seal (Figure 3B). The purpose of the stainless steel door is to minimize the electrostatic setting of particles on the polycarbonate door.

Airtight feeders and water bottles are connected to the chambers. Butterfly valves in the feeders are used for adding food without disrupting the airflow inside the chamber (Figure 4a). Bars installed inside the feeders prevent the animal from reaching the butterfly valve (Figure 4b).

A particle separator module was designed to intercept large particles by impaction while letting smaller particles flow through. The separator is composed of a stainless steel plate with 160 orifices distributed in four rows of 40 orifices (Figure 5A). Each orifice is 6.4 mm deep and 5 mm in diameter. An impaction plate is located 5 mm from the outlet of each orifice.

The velocity of the airborne particles increases as the air from cage number two is forced into cage number three through the orifices of the particle separation module by a suction pump placed upstream of cage number three. Larger particles are impacted on the impaction plates and smaller particles follow the air stream into cage number three (Figure 5b).

The pressure drop across the particle separator is recorded to ensure proper separator function. Temperature and relative humidity are also recorded. All probes are installed on the top of cages, in the inner space between cages one and two and cages two and three. The airflow can be set from 200 L/min to 400 L/min, which correspond to 12 to 25 air changes per hour. Air sampling can be programmed in the three cages, and the system airflow is adjusted automatically to maintain the efficiency of the particle separator.

### 2.2. Particle Separator D_50_ Measurement

Polydispersed aerosols were produced from Arizona road dust (ISO 12013-1, A2 fine, PTI Powder Technology Inc., Arden Hills, MN, USA) with a powder generator (fluidized bed 3400A, TSI Inc. Shoreview, MN, USA) placed inside cage number one. The powder generator was operated at 25 psi, with a bed purge of 2 L/min, a bed flow of 9 L/min and a chain rotation speed of 40. For each experiment, the aerosol generator was run for 2 h to stabilize the aerosol distribution inside the chamber. A stabilization period of 30 min was also allowed every time the flow rate of the chamber was modified. The aerodynamic distribution of the aerosol was measured with an aerodynamic particle sizer (APS) (model 3321, TSI Inc. Shoreview, MN, USA) equipped with a diluter (model 3302A, TSI Inc. Shoreview, MN, USA) using a dilution factor of 1/20. The APS and diluter were placed under cage number three and connected to a sampling port located at the bottom of the cage at 2″ from the particle separator.

The aerosol distribution was measured either with or without the particle separator between cages number two and three at flow rates of 200 L/min and 400 L/min. Measurements were also taken at 400 L/min with 75% of the orifices from the particle separator blocked with masking tape, leaving only 40 orifices open for the passage of air.

For every particle size from the APS, a mean count was calculated from 25 to 40 min of readings. The mean count obtained with the particle separator was divided by the mean count obtained at the same flow rate without the separator, thus giving a ratio of particles passing through the separator for each particle size. These ratios were plotted on graphs as illustrated in Figure 6 to estimate the D_50_ diameter. The experiment was repeated four times and the mean D_50_ diameter for each condition used was extrapolated.

### 2.3. Nano Particles Concentration in Cages Two and Three

The polydispersed nanometer particle size was generated using a collision 6-jet nebulizer (BGI, Waltham, MA, USA), filled with 50 mL of buffer (20 mM Tris-HCl, 100 mM NaCl, 10 mM MgSO4, pH 7.5) and supplied with 20 psi. Aerosols were passing through a diffusion dryer (model 3062, TSI Inc. Shoreview, MN, USA) and a neutralizer (model 3012A, TSI Inc. Shoreview, MN, USA) before entering the chamber in cage number one. Nebulization was started 30 min before starting measurements to stabilize the particle concentration in the three chambers. Measurements were performed using a NanoScan SMPS (model 3910, TSI Inc. Shoreview, MN, USA). The NanoScan sampled cage two for 10 min and then sampled cage three for 10 min. The experiments were repeated three times in all conditions. Particle size distribution as well as total nanoparticles were compared between cages two and three.

### 2.4. Experiments with Ferrets

Three groups of three seronegative (800- to 1000-*g*) male ferrets (Triple F Farms Inc., Gillett, PA, USA) were housed consecutively in the system for 7 to 12 days. The ventilation system was set at 200 L/min, with 160 holes of the particle separator for all experiments. Ferrets housed in cage one were infected intra-nasally with 250 μL (125 μL per nostril) containing 4.5 log TCID50/mL of the A/California/7/2009 (H1N1) influenza A virus. Nasal wash was collected every day by instillation of 5 mL Phosphate-Buffered Saline (PBS) into the intranasal cavity.

System ventilation was stopped before opening the cages’ sealed doors. Animals were manipulated in the following order: first the ferret from cage three, followed by the ferret from cage two, and then the ferret from cage one. Viral titer from the nasal wash was determined by plaque assay on ST6GalI-MDCK cells.

Air samples were collected every day using NIOSH two-stage bioaerosol cyclone samplers and SKC BioSamplers. Air samplers were connected to sampling ports located in cage two (between the perforated grates of cages one and two) as well as in cage three (between the particle separator and the perforated grate). Air sampling with NIOSH two-stage bioaerosol cyclone samplers was performed at 2 L/min for 24 h. At this flow rate, the cut-off separations of the NIOSH two-stage bioaerosol cyclone sampler were: 4 µm for first stage, 1.7 µm for second stage, and the remaining particles were collected on the backup filter. Air sampling started when ferrets were placed in cages after the infection of the ferret from cage one, and was stopped before shutting down the ventilation system for the daily nasal wash. Samples were eluted from NIOSH two-stage bioaerosol cyclone samplers by vortexing for 1 min in MEM (minimal essential medium; 5 mL in first stage, 500 µL in second stage, 5 ml in backup filter). Air sampling with SKC BioSamplers was performed at 11–14 L/min (determined by critical opening of the instrument) for 20 min and was set before shutting down the ventilation system for daily animal care. SKC BioSamplers were filled with 20 mL of MEM (minimal essential medium) without bovine serum albumin (BSA). After air sampling, 150 µL of BSA was added to the remaining liquid of the SKC BioSampler. Air samples were kept frozen at −80 °C until further quantitation. The virus concentration in NIOSH two-stage bioaerosol cyclone air samples was measured using qPCR [16]. The virus concentration in BioSampler air samples was measured using plaque assays on ST6GalI-MDCK cells and embryonated chicken eggs [17].

Animal procedures were approved by the Institutional Animal Care Committee of Université Laval according to the guidelines of the Canadian Council on Animal Care (protocol 2015031).

## 3. Results

### 3.1. System Validation

The D_50_ diameters at a chamber flow rate of 400 L/min were 5.07 ± 0.18 µm and 6.36 ± 0.22 µm with 40 and 160 open orifices, respectively, and 7.8 ± 0.78 µm at 200 L/min with 160 open orifices (Figure 6). To ensure correct air sampling, measurements were also made at 2″ from the air exhaust of cage number three for validation purposes. The D_50_ values measured at 2″ from air exhaust were the same (data not shown).

Data collected from two sampling ports in cage number three (2″ after the particle separator and 2″ before the exhaust) were used to document particle deposition at 200 L/min and 400 L/min without the particle separator. Particle deposition was the same at the two flow rates. The deposition of particles of particles <3.5 µm was less than 30%.

The presence or absence of the particle separator had no significant effect on the nanoparticle size distribution (10 nm to 1 µm) between cage number two and cage number three. Under all conditions the total concentration of nanoparticles in cage number three was 0% to 12% lower than that in cage number two (Table 1).

### 3.2. Experiments with Ferrets

In all three experiments, influenza A virus was detected from the index ferret (cage one) nasal wash from day 1 to days 6 or 7 (Figure 7). Nasal washes from ferrets housed in cage two were positive from days 3 or 5 in all three experiments. Influenza virus was detected from nasal washes of the ferret housed in cage three 7 days after the infection of the index ferret in experiment 1, and after 4 days in experiment 3. No virus was detected in the nasal wash of the ferret housed in cage three from experiment 2.

In all experiments, airborne influenza virus genome concentrations up to 10^4^ genomes/m^3^ were detected from cages two and three using the NIOSH two-stage bioaerosol cyclone sampler (Figure 7) from day 2 until the end of the experiment.

The influenza virus genome concentration was higher in cage two compared to cage three, except for experiment 2 on day 5. Influenza virus genomes were detected in the NIOSH backup filter in only one sample (experiment 3, cage three, day 3). No cultivable viruses were detected from SKC BioSamplers air samples using plaque assay and embryonated chicken eggs (data not shown).

## 4. Discussion

The particle separator was efficient to prevent the circulation of droplets between cages two and three, as demonstrated with dust experiments. Airflow modulation impacts the particle separator D_50_. However, airflow must be adjusted in the range of animal comfort. Therefore, the airflow should be maintained between 163 L/min and 400 L/min, which correspond to 10 and 25 air changes per hour, respectively.

Except for one sampling day where viral genomes were detected on the backup filter, influenza virus genomes were detected in the NIOSH first and/or second stages only. This means that most genomes emitted by sick ferrets were carried on particles larger than 1.7 µm. This result is consistent with the results obtained by Zhou et al. [12].

The ferrets housed in cage two were infected with influenza virus in all three experiments. This means that influenza-positive ferrets (such as the index ferret in cage one) can emit airborne particles and/or droplets containing infectious influenza virus in sufficient concentration for disease transmission without direct contact. Ferrets housed in cage three were infected with influenza virus in two out of three experiments. This result indicates that ferrets can be infected by influenza virus carried on airborne particles emitted by influenza positive ferrets, in accordance with the literature [12,14].

Nasal washes of ferrets housed in cage two were positive 2–4 days after the infection of ferrets housed in cage one. Nasal washes of ferrets housed in cage three were positive 1–4 days after washes from ferrets housed in cage two were found to be positive for influenza. The delay between the influenza detection schedule in ferrets in cages two and three can be explained by a lower virus concentration in the air and the infection route. Indeed, the airborne influenza virus genome concentration was lower in cage three compared to cage two in 19 out of 20 sampling days. Large particles eliminated by the separator likely contained high virus concentrations. The airborne influenza virus genome concentration in cage three reached 5 × 10^2^ genomes/m^3^ only when influenza virus was detected in the nasal wash of the ferret housed in cage two. Therefore, it is possible that the airborne virus concentration in cage three reached the required concentration to transmit the infection only when the ferret housed in cage two showed flu symptoms. More experiments would be required to elucidate this phenomenon.

## 5. Conclusions

This paper describes a chamber system that can be used for airborne disease transmission studies. The system is airtight and particle size distribution through the system is satisfactory (no significant difference was found for particles <420 nm). The particle separator D_50_ can be modulated between 5.07 and 7.8 µm by changing the airflow. To further reduce the D_50_, plates with smaller holes could replace the actual hole plates of the particle separator. This would lead to more accelerated particles and, therefore, more particles being captured by the impaction plates. Ferrets can be used as models for the study of many mammalian viruses, including filovirus [18], respiratory syncytial virus [19] and Morbillivirus [20]. Therefore, the studies that can be conducted in this chamber system are not limited to influenza viruses. Moreover, this cage system can be adapted to accommodate other animals like rats or rabbits. Indeed, smaller, regular cages can be placed inside the ventilated airtight cages.

## Figures and Tables

**Figure 1 ijerph-16-00609-f001:**
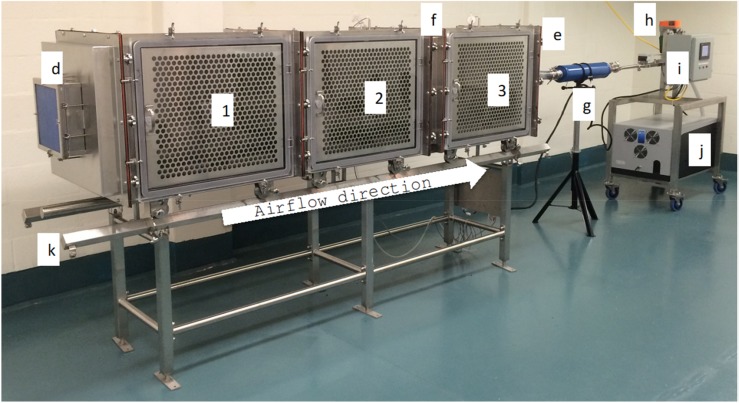
System overview. (**1**) Cage number one, (**2**) cage number two, (**3**) cage number three, (**d**) high-efficiency particulate air (HEPA) filter inlet air, (**e**) HEPA filter exhaust air (not visible in the picture), (**f**) particle separator, (**g**) muffler, (**h**) airflow adjustment valve, (**i**) control panel, (**j**) pump installed in an insulated box, (**k**) support table with rails.

**Figure 2 ijerph-16-00609-f002:**
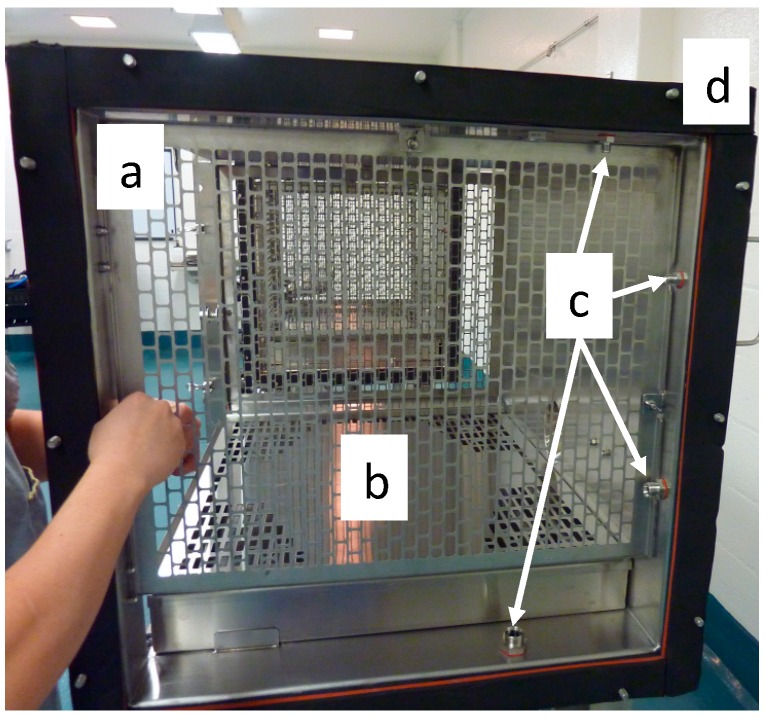
Side view of a cage. (**a**) Perforated grates on each side of the cage, (**b**) excreta pan with a perforated lid, (**c**) sampling ports, (**d**) rubber seal between cages.

**Figure 3 ijerph-16-00609-f003:**
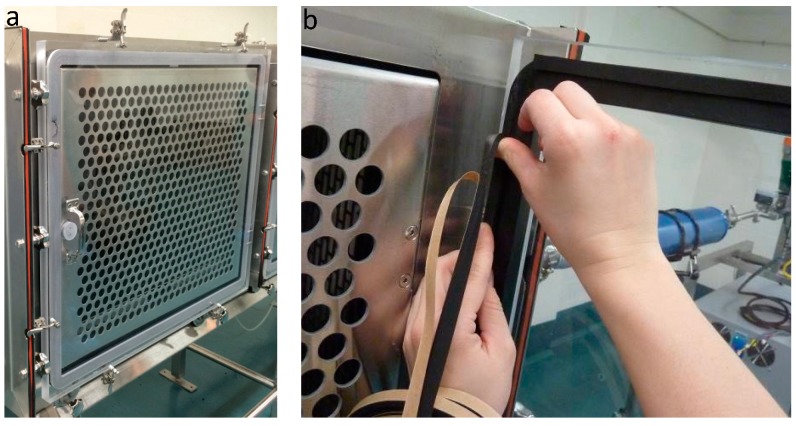
Cage door components. (**a**) Perforated stainless steel door and transparent polycarbonate door. (**b**) Rubber seal of the polycarbonate door.

**Figure 4 ijerph-16-00609-f004:**
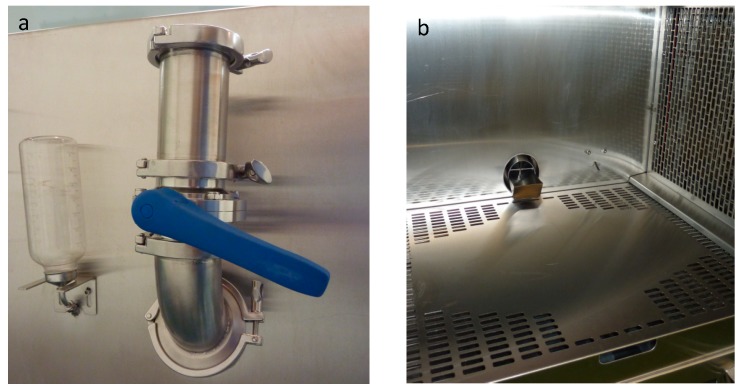
Feeder and water bottle. (**a**) Outside view of the feeder with butterfly valve and water bottle installed on a cage, (**b**) cage inside view with feeder, animal water supply and excreta pan.

**Figure 5 ijerph-16-00609-f005:**
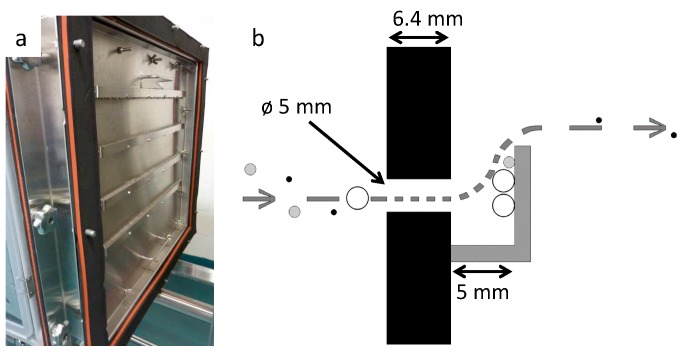
Particles separator. (**a**) Assembly of stainless steel plate with four rows of 40 orifices. On the picture, orifices are covered with impaction plates located 5 mm from the orifice’s outlets. (**b**) Schematic representation of the particle separator principle and design.

**Figure 6 ijerph-16-00609-f006:**
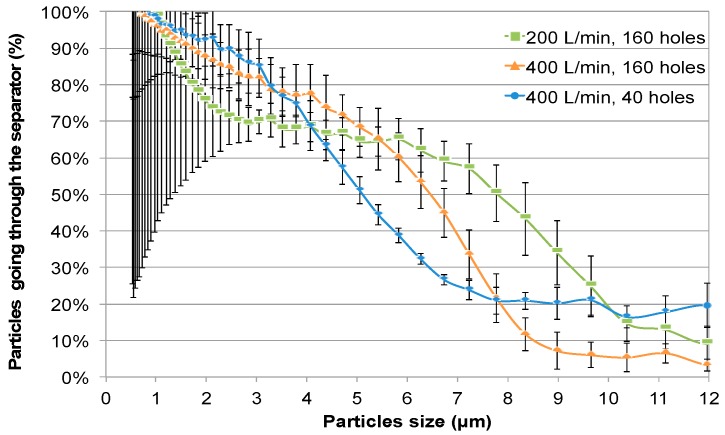
Particle separator D_50_ measurement as a function of airflow and the number of separator orifices used. Comparison of particle distribution in cage three with and without a particle separator, as measured with an aerodynamic particle sizer (APS) located at 2″ from the particle separator.

**Figure 7 ijerph-16-00609-f007:**
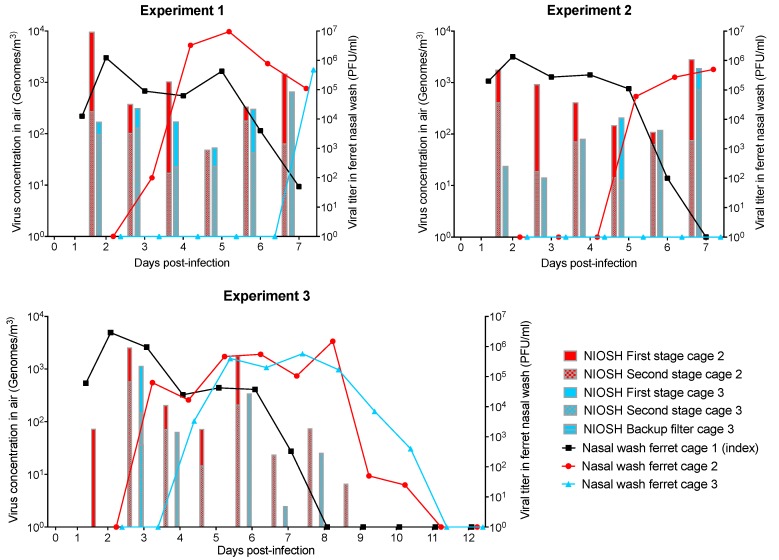
Influenza genome per cubic meter of air, and influenza virus titer in nasal washes of ferrets hosted in cage system for 7 or 12 days from three experiments. The index ferret (cage one) was infected on day 0. Air samples were collected using National Institute for Occupational Safety and Health (NIOSH) two-stage bioaerosol cyclone samplers. Genome concentrations found with the NIOSH first stage, second stage and backup filter are superimposed.

**Table 1 ijerph-16-00609-t001:** Nanoparticles (10 nm–420 nm) total concentration difference between cages two and three.

Airflow—Particles Separator Configuration	Decrease in Concentration
200 L/min—no separator	0% ± 9%
200 L/min—160 orifices	11% ± 5%
400 L/min—no separator	12% ± 19%
400 L/min—160 orifices	6% ± 5%
400 L/min—40 orifices	11% ± 5%

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
