# Peer review of "Design and Validation with Influenza A Virus of an Aerosol Transmission Chamber for Ferrets"

_ijerph, 2019, doi:10.3390/ijerph16040609_

Reviewer 1 Report

The authors constructed air-tight cages and put 3 of these cages in a row with a controlled amount of clean air (HEPA filter) entering the first cage, then passing on to the second and then to the third cage. Between 2nd and the 3rd cage they placed a particle separator working according to the impactor principle. With the empty cages they tested the efficiency and cut-off level of the separator. It stands to reason that the higher the velocity on the impactor the smaller the particles that get separated. Indeed the cut-off diameter was smaller with higher air exchange flow, but closing 120 of the 160 holes did not reduce the cut-off diameter much further. A doubling of the flow rate (200 versus 400 L/min) reduced the cut-off diameter from 7.8 to 6.4 µm. A further 4-fold increase of the flow velocity (by reducing the number of holes to 25%) only reduced the cut-off diameter further from 6.4 to 5.1 µm. We do not know how the separator will work when animals are housed in the cages. Animals will move around and will thus likely increase turbidity of the air flow. That could affect the separation efficiency and cut-off diameter of the separator.

The main experiment was about virus transmission between animals. Influenza A virus was used as a model virus and ferrets were used (in 3 consecutive experiments) as model animals. We learn about sampling volume but not about flow rate during these 3 experiments. Just 3 experiments might be not sufficient to draw clear conclusions (e.g. if a cut-off diameter of 5 µm reduces infectivity compared to one of 8 µm). And as explained above even if the flow rates were provided we cannot be sure that the cut-off diameters would have been the same as with empty cages.

The experiments indicate that the ferret in the first cage does infect the ferrets in the consecutive cages. Infection of the third animal might have occurred indirectly with virus transfer via the second ferret. But at least in one case the latency period between the start of the infection in the second and the third animal was only one day which is maybe too short considering the usual incubation period.

Viral DNA was detected in the air of all cages with higher virus load in cage 2 compared to cage 3. Viruses were detected in each cage well before infection was present in the respective animal. But infection of the animal again boosted virus load. All this is not very surprising. Air samples were collected by NIOSH two-stage filters so virus load in larger and smaller particles (4 µm and 1.7 µm cut-off respectively) could be analysed separately.

The paper is clear and straightforward. Apart from the issue about the true cut-off of the separator in the animal experiments all procedures are well described. Only I am not sure about the practical use of this experimental system. What would be an interesting research question to ask that could be answered? Viruses causing airway infections are bound to be transported in the air. This is their mode of operation really! Showing that it works is not very surprising. We might want to know about the necessary dose to cause infection. We might wonder how long virus material stays infective in the air given various environmental conditions (humidity, temperature, UV-radiation, ozone concentration,…). But are animals a suitable model for human infection? Animals differ in their sneezing mechanism thus producing droplets of a size different from humans. Animals differ in the anatomy of their respiratory tract. So sedimentation of particles by size differs from humans. Also infectivity for various viruses differs by species. All this considered I find the presented experiments only inform us about the infection risk of ferrets in a highly specialised environment. Even the cut-off diameter (around 5-7 µm) might not be very relevant for most practical purposes.

Minor points: The authors differentiate between “aerosol” and “droplets”. Their differentiation seems to be arbitrarily based on the cut-off diameter of their separator. In physics (and in air pollution research) an “aerosol” is a suspension of liquid and solid particles in a gas, usually in air. Both liquid and solid particles can have different diameters. With viral airway infections droplets of different size (size range depending on species) are produced through sneezing and coughing. In these droplets the virus particles are protected from environmental impacts. In dry air part of the droplets will evaporate. Droplets will fall down and attach to surfaces and will dry there. Smaller particles might come re-suspended into the air with mechanical forces (e.g. movement of the animal) acting on the surfaces. Eventually we will find virus DNA in particles of different size. The separation in droplets and particles (with a cut-off at 5 µm) is neither in line with physical nomenclature nor does it make much sense in relation to virology.

Line 60: a “reference” is mentioned (“[REF]”) but not provided.

Line 143: “The APS and diluter were placed under cage number three…” We are not so much interested where the instruments are placed but where the samples are taken from. I assume a sample outlet is located at the bottom of the cage?

Line 212: m³, not m3!

Line 227: “and/or” is not very precise. In fact virus DNA was detected in both stages. So “and” would be correct. I am not so sure where a third stage was analysed with a two-stage sampler. And only here in the discussion do we learn that in this 3rd stage no virus was detected. Maybe this refers to the “backup filter” mentioned in the results and in figure 7. But this backup filter is not mentioned in the methods and in fact at least on one day virus was detected on this filter.

Line 249: “satisfactory” asks for an explanation: satisfactory for what purpose?

Line 251: This sentence could be improved grammatically: “hole plates of the particle separator could be replaced by plated with smaller holes”

Author Response

We would like to tank the reviewers for the insightful comments. They raised very good points and noticed some aspects that were not clear or that have been forgotten. We thank them for their precious time and their careful review.

Reviewer 1

The authors constructed air-tight cages and put 3 of these cages in a row with a controlled amount of clean air (HEPA filter) entering the first cage, then passing on to the second and then to the third cage. Between 2nd and the 3rd cage they placed a particle separator working according to the impactor principle. With the empty cages they tested m air exchange flow, but closing 120 of the 160 holes did not reduce the cut-off diameter much further. A doubling of the flow rate (200 versus 400 L/min) reduced the cut-off diameter from 7.8 to 6.4 µm. A further 4-fold increase of the flow velocity (by reducing the number of holes to 25%) only reduced the cut-off diameter further from 6.4 to 5.1 µm. We do not know how the separator will work when animals are housed in the cages. Animals will move around and will thus likely increase turbidity of the air flow. That could affect the separation efficiency and cut-off diameter of the separator.

The main experiment was about virus transmission between animals. Influenza A virus was used as a model virus and ferrets were used (in 3 consecutive experiments) as model animals. We learn about sampling volume but not about flow rate during these 3 experiments. Just 3 experiments might be not sufficient to draw clear conclusions (e.g. if a cut-off diameter of 5 µm reduces infectivity compared to one of 8 µm). And as explained above even if the flow rates were provided we cannot be sure that the cut-off diameters would have been the same as with empty cages.

The experiments indicate that the ferret in the first cage does infect the ferrets in the consecutive cages. Infection of the third animal might have occurred indirectly with virus transfer via the second ferret. But at least in one case the latency period between the start of the infection in the second and the third animal was only one day which is maybe too short considering the usual incubation period.

Viral DNA was detected in the air of all cages with higher virus load in cage 2 compared to cage 3. Viruses were detected in each cage well before infection was present in the respective animal. But infection of the animal again boosted virus load. All this is not very surprising. Air samples were collected by NIOSH two-stage filters so virus load in larger and smaller particles (4 µm and 1.7 µm cut-off respectively) could be analysed separately.

The paper is clear and straightforward. Apart from the issue about the true cut-off of the separator in the animal experiments all procedures are well described. Only I am not sure about the practical use of this experimental system. What would be an interesting research question to ask that could be answered? Viruses causing airway infections are bound to be transported in the air. This is their mode of operation really! Showing that it works is not very surprising. We might want to know about the necessary dose to cause infection. We might wonder how long virus material stays infective in the air given various environmental conditions (humidity, temperature, UV-radiation, ozone concentration,…). But are animals a suitable model for human infection? Animals differ in their sneezing mechanism thus producing droplets of a size different from humans. Animals differ in the anatomy of their respiratory tract. So sedimentation of particles by size differs from humans. Also infectivity for various viruses differs by species. All this considered I find the presented experiments only inform us about the infection risk of ferrets in a highly specialised environment. Even the cut-off diameter (around 5-7 µm) might not be very relevant for most practical purposes.

Response: based on the reviewer comments, we realized that our goal was not clearly explained. We added several information and references in the introduction to better explain the purpose of the system.

Minor points: The authors differentiate between “aerosol” and “droplets”. Their differentiation seems to be arbitrarily based on the cut-off diameter of their separator. In physics (and in air pollution research) an “aerosol” is a suspension of liquid and solid particles in a gas, usually in air. Both liquid and solid particles can have different diameters. With viral airway infections droplets of different size (size range depending on species) are produced through sneezing and coughing. In these droplets the virus particles are protected from environmental impacts. In dry air part of the droplets will evaporate. Droplets will fall down and attach to surfaces and will dry there. Smaller particles might come re-suspended into the air with mechanical forces (e.g. movement of the animal) acting on the surfaces. Eventually we will find virus DNA in particles of different size. The separation in droplets and particles (with a cut-off at 5 µm) is neither in line with physical nomenclature nor does it make much sense in relation to virology.

Response: We agree with the reviewer that the delimitation between aerosols and droplets is arbitrary since it depends of the fate of the aerosols once they are produced. In natural environment, liquid from droplets can evaporate and give droplet nuclei with much smaller diameter and can remain airborne for a long time. Other droplets can settle before liquid evaporation. The fate of aerosols depend on many factors. Since the purpose of the manuscript was not to explain the concepts of droplets and droplets nuclei, we therefore replaced the term “droplet nuclei” from the manuscript by “aerosols”. For reader comprehension, we decided to stick to the World Health Organisation of droplet and aerosols transmission (particles > 5µm are droplets and particles < 5µm are aerosols). We added the definition and the reference in the introduction.

Line 60: a “reference” is mentioned (“[REF]”) but not provided.

Response: Agree, reference was added (Zhou, J., Wei, J., Choy, K.T., Sia, S.F., Rowlands, D.K., Yu, D., Wu, C.Y., Lindsley, W.G., Cowling, B.J., McDevitt, J., Peiris, M., Li, Y., and Yen, H.L. Defining the sizes of airborne particles that mediate influenza transmission in ferrets. Proc. Natl. Acad. Sci. U S A 2018, 115, E2386-E2392.)

Line 143: “The APS and diluter were placed under cage number three…” We are not so much interested where the instruments are placed but where the samples are taken from. I assume a sample outlet is located at the bottom of the cage?

Response: Yes, exactly, the sampling port is located on the bottom of the cage. This information was added in the text.

Line 212: m³, not m3!

Response: Agree, text modified accordingly.

Line 227: “and/or” is not very precise. In fact virus DNA was detected in both stages. So “and” would be correct. I am not so sure where a third stage was analysed with a two-stage sampler. And only here in the discussion do we learn that in this 3rd stage no virus was detected. Maybe this refers to the “backup filter” mentioned in the results and in figure 7. But this backup filter is not mentioned in the methods and in fact at least on one day virus was detected on this filter.

Response: We used and/or because viral genome were not detected on both stages at every sampling day. Sometimes it was on the first stage, sometimes on the second stage and most of the times on both. The reviewer is correct: the third stage is the backup filter. We added the following information in the materiel and methods section to avoid readers confusion:  “Air sampling with NIOSH air samplers was performed at 2 L/min for 24 h. At this flowrate, the cut-off separation of the NIOSH air sampler is: 4 µm for first stage, 1.7 µm for second stage, and remaining particles are collected on the backup filter.” We agree with reviewer that the sentence in line 227 was confusing. We modified the text as followed: “Except for one sampling day where viral genomes were detected on the backup filter, influenza virus genomes were detected on NIOSH first and/or second stage only. It means that most genomes emitted by sick ferrets are carried on particles larger than 1.7 µm. This result is consistent with results obtained by Zhou et al. [6].”

Line 249: “satisfactory” asks for an explanation: satisfactory for what purpose?

Response: It is satisfactory for particles < 420 nm. We added this precision in the text.

Line 251: This sentence could be improved grammatically: “hole plates of the particle separator could be replaced by plated with smaller holes”

Response: Agree, sentence modified as follow:To reduce further the D50, plates with smaller holes could replace the actual hole plates of the particle separator. It will lead to more accelerated particles and therefore, more particles captured by the impaction plates.”

Reviewer 2 Report

This paper describes the design and validation of a three-compartment chamber for studying transmission of viruses by aerosol particles.  The design included an impactor to remove particles larger than 5 to 8 microns between the second and third chambers.  The chamber was tested with laboratory generated particles and with ferrets.  While the authors clearly put a lot of effort into building and testing the chamber, it is not clear what the point is.  Previous research does not show much difference in virus transmission for particles above and below 5 to 8 microns (e.g., Zhou et al. 2018).  Why did the authors choose this size as the cut point?  It is also not clear why the authors chose a three-compartment system.  What new information does the third compartment provide?

Page 1, lines 36-37 and elsewhere.  I don’t understand the distinction the authors are making between droplets and droplet nuclei.  Droplet nuclei suggests that the droplets have been dried to give residual particles, but I don’t think they are drying the particles in the ferret experiments.  On page 2, line 81, it looks like they define the difference between droplets and droplet nuclei by the cutoff size in the impactor, but it is unlikely that there is any difference in composition or water content above and below this size.  It would be better if they stick with the definitions on page 2, line 45 for droplets (5 to 15 microns) and particles (1.5 to 5 microns), though even that is somewhat arbitrary. 

Page 2, lines 54-56.  This sentence is confusing.  Please revise.

Page 2, line 60. Reference is missing.

Page 2, line 73.  How does the separator prevent particle settling? 

Page 4, line 117.  The text and Figure 5 are different.  In the Figure, it looks like the orifice is 6.4 mm in diameter and 5 mm deep.

Page 4, line 115.  How did the authors design the impactor?  Did they use modeling?  Does the measured size cut agree with the model?  Can they change the distance to the impaction plate to change the size cut? 

Page 5, line 134.  This section is really about measuring the particle separator D50, not calculating it.  They don’t present any results from a computational model.

Page 5, line 143.  The authors say that the APS and diluter were placed underneath cage three.  Were they sampling through the port at the bottom of cage three right after the separator?  Do they know that cage three is well-mixed?  This sampling location could be problematic if the chamber is not well-mixed and there is high air flow.  Ideally, one would want to sample from the middle of cage three.  Did they measure the size distribution in cage 2? Is it the same as in cage three without the separator?  Particles larger than a few microns are notoriously hard to sample without distorting the size distribution.

Page 6, line 175.  What are the size cuts on the stages of the NIOSH sampler?  What liquid did the authors use in the SKC Biosampler?  It would be good to include references for both of these devices.

Page 6, Figure 6.  This is a measurement of D50, not a calculation.  Typically, one plots particle diameters on a log scale because most particle size distributions are log normal.  These are rather gradual cutoffs. Did they grease the impaction plate?  A thin layer of grease can reduce bounce and improve the sharpness of the size cut.

Page 6, line 196.  The size range of the Nanoscan is 10 to 420 nm.  Please correct here and in Table 1.  What was the point of this measurement?  If the particle separator is designed to have a cut point of several microns, it should have no impact on sub-micron particles.  Were the same sampling ports used for the Nanoscan as for the APS?  Did the authors measure the particles in cage 1?  Determining low particle losses for particles less than 0.5 microns does not say anything about possible losses for particles greater than a few microns.

Page 6, Table 1 column headers. Concentration difference is ambiguous.  Please indicate that these are decreases in concentration.

Page 7, line 202.  What are the three experiments?  Were flow conditions the same?  Number of orifices in the separator?  Can that explain the differences in how quickly ferret 2 and 3 were infected (or not infected)?

Page 7, Figure 7.  It is very difficult to tell the difference between stage one and stage two in the bars.  Please use more contrast.  NIOSH is spelled incorrectly two different ways in the caption.

Page 7, lines 214-215.  What are the error bars on the measurements of genome concentration?  I suspect that cage 2 and cage 3 are actually the same within the error bars for many of these measurements.

Page 8, lines 241-242.  The authors state “Large particles eliminated by the separator likely contain high virus concentration.”  This contradicts the statement on page 7, lines 225-226 that removing the large particles should not change the genome concentration in the NIOSH collector. 

Did the authors measure size distributions for the particles generated by the ferrets?  There are probably very few particles above 5 to 8 microns, so I would guess that the particle separator has very little impact on the ferret generated size distributions in cage 2 and cage 3.

From the time delay between ferrets in Figure 7, it looks like ferret 1 infects ferret 2 and then ferret 2 infects ferret 3.  If ferret 1 is not directly infecting ferret 3, that suggests that large virus-containing particles are not getting through cage 2 into cage 3.  This doesn’t seem consistent with the authors claim that particle transmission is good across the whole size range.  During the dust experiments, did the authors measure the size distribution in cage 2? Or at the particle generator?  These measurements would give a better characterization of particle transmission in the size range relevant for virus transmission.

Author Response

We would like to tank the reviewers for the insightful comments. They raised very good points and noticed some aspects that were not clear or that have been forgotten. We thank them for their precious time and their careful review.

Reviewer 2

This paper describes the design and validation of a three-compartment chamber for studying transmission of viruses by aerosol particles.  The design included an impactor to remove particles larger than 5 to 8 microns between the second and third chambers.  The chamber was tested with laboratory generated particles and with ferrets.  While the authors clearly put a lot of effort into building and testing the chamber, it is not clear what the point is.  Previous research does not show much difference in virus transmission for particles above and below 5 to 8 microns (e.g., Zhou et al. 2018).  Why did the authors choose this size as the cut point?  It is also not clear why the authors chose a three-compartment system.  What new information does the third compartment provide?

Response: based on the reviewer comments, we realized that our goal was not clearly explained. We added several information and references in the introduction to better explain the purpose of the system.

Page 1, lines 36-37 and elsewhere.  I don’t understand the distinction the authors are making between droplets and droplet nuclei.  Droplet nuclei suggests that the droplets have been dried to give residual particles, but I don’t think they are drying the particles in the ferret experiments.  On page 2, line 81, it looks like they define the difference between droplets and droplet nuclei by the cutoff size in the impactor, but it is unlikely that there is any difference in composition or water content above and below this size.  It would be better if they stick with the definitions on page 2, line 45 for droplets (5 to 15 microns) and particles (1.5 to 5 microns), though even that is somewhat arbitrary. 

Response: To avoid readers’ confusion, we replaced the term droplet nuclei by aerosols in the manuscript. We also add the World Health Organisation definition of droplets vs aerosol transmission (particles > 5µm are droplets and particles < 5µm are aerosols).

Page 2, lines 54-56.  This sentence is confusing.  Please revise.

Response: We revised the sentence as followed: “By exposing animals to airborne viruses, air sampling can be avoided (preventing virus damage) as well as laboratory virus culture bias in detection.”

Page 2, line 60. Reference is missing.

Response: Agree, reference was added (Zhou, J., Wei, J., Choy, K.T., Sia, S.F., Rowlands, D.K., Yu, D., Wu, C.Y., Lindsley, W.G., Cowling, B.J., McDevitt, J., Peiris, M., Li, Y., and Yen, H.L. Defining the sizes of airborne particles that mediate influenza transmission in ferrets. Proc. Natl. Acad. Sci. U S A 2018, 115, E2386-E2392.)

Page 2, line 73.  How does the separator prevent particle settling? 

Response: the sentence was confusing. We modified as followed:” We designed a particle separator to prevent large droplets transmission between cages.”

Page 4, line 117.  The text and Figure 5 are different.  In the Figure, it looks like the orifice is 6.4 mm in diameter and 5 mm deep.

Response: Agree, the correct information was in the text. We modified the figure accordingly.

Page 4, line 115.  How did the authors design the impactor?  Did they use modeling?  Does the measured size cut agree with the model?  Can they change the distance to the impaction plate to change the size cut? 

Response: No, we did not use modelling. We tested a prototype of 8 holes, and we enlarged the same design to 160 holes. We can change the impaction plate in order to change the size cut, but we have not tried it.

Page 5, line 134.  This section is really about measuring the particle separator D50, not calculating it.  They don’t present any results from a computational model.

Response: Agree, text modified accordingly.

Page 5, line 143.  The authors say that the APS and diluter were placed underneath cage three.  Were they sampling through the port at the bottom of cage three right after the separator?  Do they know that cage three is well-mixed?  This sampling location could be problematic if the chamber is not well-mixed and there is high air flow.  Ideally, one would want to sample from the middle of cage three.  Did they measure the size distribution in cage 2? Is it the same as in cage three without the separator?  Particles larger than a few microns are notoriously hard to sample without distorting the size distribution.

Response: Yes, exactly, the sampling port is located on the bottom of the cage. This information was added in the text. We also took measurements in two locations in cage 3 (2” after the separator and 2” before the exhaust) for the D50 measurement. The measurement taken at the two locations give the same D50. In the manuscript, we only display the results for the sampling port located 2” after particle separator because we have more replicates at this location so we can display error bars. We added the information about the APS location in the Material and Methods section as well as in the figure legend. We also added the information about the alternative sampling port in the result section. The MMAD of Arizona road dust (fine grade) is 9.1 µm. Therefore, we were expecting particle deposition in the cages during dust experiments. That is the reason of the measurements with smaller particles. Since small particles are not expected to settle in theses conditions, we wanted to make sure of a uniform particle distribution in the system. Difference of particle size distribution and concentration could have meant inefficient sampling due to turbulence or incorrect air mixing. We have collected data about particle deposition in cage 3 between the two sampling ports (2” after particle separator and 2” before air exhaust) without particle separator. Particle deposition was the same at the two flowrates. Deposition of particles of particles < 3.5 µm was less than 30%. We add this information in the text.

Page 6, line 175.  What are the size cuts on the stages of the NIOSH sampler?  What liquid did the authors use in the SKC Biosampler?  It would be good to include references for both of these devices.

Response: We agree with reviewer, some information was missing about air sampling and air samples processing. We add the following information in the materiel and methods section:  “Air sampling with NIOSH air samplers was performed at 2 L/min for 24 h. At this flowrate, the cut-off separation of the NIOSH air sampler are: 4 µm for first stage, 1.7 µm for second stage, and remaining particles are collected on the backup filter. Air sampling started when ferrets were placed in cages after infection of the ferret from cage 1, and was stopped before shutting down the ventilation system for the daily nasal wash. Samples were eluted from NIOSH air samplers by vortexing 1 min in MEM (5 ml in first stage, 500 µl in second stage, 5 ml in backup filter). Air sampling with SKC BioSamplers was performed at 11-14 L/min (determined by critical opening of the instrument) for 20 min and was set also before shutting down the ventilation system for daily animal care. SKC BioSamplers were filled with 20 ml of MEM (minimal essential medium) without BSA. After air sampling, 150 µl of BSA were added to the SKC BioSampler remaining liquid. Air samples were kept frozen at -80C until further quantitation.’’

Page 6, Figure 6.  This is a measurement of D50, not a calculation.  Typically, one plots particle diameters on a log scale because most particle size distributions are log normal.  These are rather gradual cutoffs. Did they grease the impaction plate?  A thin layer of grease can reduce bounce and improve the sharpness of the size cut.

Response: Agree, we modified the text accordingly. No, we did not use grease. It would have been a great idea to do so.

Page 6, line 196.  The size range of the Nanoscan is 10 to 420 nm.  Please correct here and in Table 1.  What was the point of this measurement?  If the particle separator is designed to have a cut point of several microns, it should have no impact on sub-micron particles.  Were the same sampling ports used for the Nanoscan as for the APS?  Did the authors measure the particles in cage 1?  Determining low particle losses for particles less than 0.5 microns does not say anything about possible losses for particles greater than a few microns.

Response: Thanks to the reviewer for noticing this mistake. We corrected the Table 1. Since small particles are not expected to settle in theses conditions, we wanted to make sure of a uniform particles distribution in the system. Difference of particle size distribution and concentration for particles smaller than 0.5 µm could have mean inefficient sampling due to turbulence or incorrect air mixing.

Page 6, Table 1 column headers. Concentration difference is ambiguous.  Please indicate that these are decreases in concentration.

Response: Agree, Table 1 header modified accordingly.

Page 7, line 202.  What are the three experiments?  Were flow conditions the same?  Number of orifices in the separator?  Can that explain the differences in how quickly ferret 2 and 3 were infected (or not infected)?

Response: Our apologies to the reviewers, we forgot to add this information in the manuscript that was submitted. We added this information in the Material and Methods section of this revised version. All experiments were conducted with the same flowrate and particle separator configuration.Three groups of three seronegative (800- to 1000-g) male ferrets (Triple F Farm Inc., PA) were housed consecutively in the system for 7 to 12 days. Ventilation system was set at 200 L/min, with 160 holes of the particle separator for all experiments. Ferrets housed in cage 1 were infected intra-nasally with 250 μl (125 μl per nostril) containing 4·5 log TCID50/ml of the A/California/7/2009 (H1N1) influenza A virus.

Page 7, Figure 7.  It is very difficult to tell the difference between stage one and stage two in the bars.  Please use more contrast.  NIOSH is spelled incorrectly two different ways in the caption.

Response: Agree, NIOSH spelling was corrected in the caption. We change the filing pattern in the figure to increase contrast.

Page 7, lines 214-215.  What are the error bars on the measurements of genome concentration?  I suspect that cage 2 and cage 3 are actually the same within the error bars for many of these measurements.

Response: There is no error bar since these are single measurements. Since the results were different between the 3 experiments because of the lag time between infection of ferret 1 and 2 in experiment 2, we preferred to display the 3 experiments separately instead of one graph average with error bars of the 3 experiments.

Page 8, lines 241-242.  The authors state “Large particles eliminated by the separator likely contain high virus concentration.”  This contradicts the statement on page 7, lines 225-226 that removing the large particles should not change the genome concentration in the NIOSH collector. 

Response: We agree with the reviewer that the statement on page 7 lines 225-226 was very confusing. We removed it from the manuscript.

Did the authors measure size distributions for the particles generated by the ferrets?  There are probably very few particles above 5 to 8 microns, so I would guess that the particle separator has very little impact on the ferret generated size distributions in cage 2 and cage 3.

Response: No, we did not use particle counters with ferrets.

From the time delay between ferrets in Figure 7, it looks like ferret 1 infects ferret 2 and then ferret 2 infects ferret 3.  If ferret 1 is not directly infecting ferret 3, that suggests that large virus-containing particles are not getting through cage 2 into cage 3.  This doesn’t seem consistent with the authors claim that particle transmission is good across the whole size range.  During the dust experiments, did the authors measure the size distribution in cage 2? Or at the particle generator?  These measurements would give a better characterization of particle transmission in the size range relevant for virus transmission.

Response: We modified the discussion section and hope it is clearer now. We measured the particle size distribution at two different locations in cage 3 during dust experiments. We have collected data about particle deposition in cage 3 between the two sampling ports (2” after particle separator and 2” before air exhaust) without particle separator. The deposition was the same at 200 L/min and 400 L/min. Deposition of particles of particles < 3.5 µm was less than 30%.  We add this information in the text.